# Tab2Gan: Utilizing image conversion and Gan inversion for tabular model robustness

## Abstract

New advanced adversarial attacks are emerging rapidly. These threats have prompted the development of various defense strategies, including robustness techniques. In this paper we propose a novel attack-agnostic robustness method that utilizes the generative capabilities of image based generative adversarial networks (GANS) to enhance the robustness of classical machine learning models trained on structured (tabular) data. To safeguard the target models, we employ two GANs, each trained on benign data from different classes. These GANs function as a defensive mechanism by classifying incoming inputs, whether they are benign or adversarial, and then reconstructing them within the benign data distribution of each respective GAN before presenting them to the target model. In our evaluation, conducted on three commonly known structured datasets and two conventional machine learning models, our proposed robustness approach consistently outperformed the existing techniques in the field. In most of the experiments applying the suggested robustness method yields classification accuracy results that closely align with the benign accuracy rate of 99% i.e, the target model performs as if it has not been subjected to any attack whatsoever.

## 1 Introduction

The landscape of machine learning and artificial intelligence has seen remarkable progress in recent years, accompanied by the advent of advanced adversarial attacks that pose significant threats to model performance and reliability (Chakraborty et al., 2021). These adversarial threats have necessitated the development of robustness techniques and defensive strategies to safeguard machine learning models against malicious perturbations and ensure their resilience in real-world applications.

Adversarial examples, initially introduced by Szegedy et al. (2013), represent a significant challenge in the field of machine learning (ML). These are input samples for ML models that have been carefully crafted by introducing minor, deliberate perturbations, which are designed to manipulate the model's classification outcomes. The discovery of adversarial examples has shed light on the vulnerability of ML models to malicious manipulations (Chakraborty et al., 2021). While many early adversarial attacks focused on attacking models trained on homogeneous data (Chakraborty et al., 2021), there has been a recent surge in the development of attacks specifically tailored for the intricacies of tabular data (Grolman et al., 2022).

In response to these challenges, this paper introduces an innovative and attack-agnostic approach to enhance the robustness of classical machine learning models when applied to structured (tabular) data. Leveraging the generative capabilities of image-based Generative Adversarial Networks (GANs) alongside with the GAN Inversion (Xia et al., 2022) and Tab2Img 1 conversion techniques, our proposed method represents a novel paradigm in defending against adversarial attacks in domains typically less explored by GAN-based defenses.

Our approach capitalizes on the versatility of GANs to not only detect adversarial inputs but also reconstruct them within the benign data distribution, effectively mitigating the impact of potential attacks. By employing two GANs, each trained on benign data originating from distinct classes, we create a potent defense mechanism that exhibits remarkable adaptability in classifying both benign and adversarial inputs.

In the subsequent sections, we provide a comprehensive description of our proposed attack-agnostic robustness method, detailing the architecture of the framework and training of the GANs, their roles as defensive shields for target models, and the evaluation process conducted on diverse structured datasets. Our experimental results demonstrate the remarkable efficacy of our approach, consistently surpassing the performance of existing robustness techniques in the field. Notably, our approach achieves classification accuracy rates closely aligned with the benign accuracy, ensuring that the target model retains its high-performance standards even in the presence of adversarial challenges.

Recently it has been shown that applying image conversion techniques to convert tabular records into images has great potential in many domains (Arik & Pfister, 2021; Sharma et al., 2019). Our main contribution is that to the best of our knowledge we are the first to apply this technique to improve the robustness of tabular data based models against adversarial attacks. Secondary contribution being, that we show GANs can find spacial correlations and even reconstruct them in images that were transformed from tabular records.

## 2 BACKGROUND

### 2.1 TAB2IMG AND IMAGE RESIZING

Sharma et al. (2019) proposed a method for converting tabular data to image data. The method was tested in different domains such as medical (RNA-seq), vowels, text, and artificial data. Consequently, a python package named Tab2Img[1] was created with the intent of providing a general methodology to convert any non-image data to an image. Given a training dataset $X$ with $m$ samples and $d$ features, Tab2Img finds a function that maps the tabular records onto $nxn$ images where $n = \lceil \sqrt{d} \rceil$. This is done using the Pearson correlation coefficient between $X$ and the target vector $Y$, sorting it from greatest to smallest values which generates a vector of indices, and then ordering the indices in a $nxn$ matrix which will later be used to map tabular records to images.

Image resizing stands as a pivotal operation within the domain of digital image processing, with the Python Imaging Library, now commonly referred to as Pillow Clark, 2015, emerging as a cornerstone tool in simplifying this intricate task. The profound significance of PIL's image resizing capabilities becomes apparent as researchers and developers delve into the complexities of image manipulation. This library empowers us to efficiently alter image dimensions while retaining the intrinsic visual essence.

### 2.2 GENERATIVE ADVERSARIAL NETWORKS AND THE INVERSION OF THEIR GENERATOR

Generative Adversarial Networks (GANs) are mostly applicable across various computer vision tasks, including but not limited to image translation, image manipulation, and image restoration (Xia et al., 2022). The GAN proposed by Goodfellow et al. (2014) is a deep generative model designed to acquire the skill of producing novel data via an adversarial training approach. This framework consists of a pair of neural network components: one known as the generator (G) and the other as the discriminator (D) which are trained collaboratively through an adversarial process. The primary goal of the generator (G) is to create synthetic data that closely resembles genuine data, whereas the discriminator (D) aims to differentiate between genuine and synthetic data. Through this training process, the generator (G) endeavors to generate synthetic data that convincingly mimics the distribution of real data, trying to deceive the discriminator, while never actually receiving the real data (Goodfellow et al., 2014).

The generator of a GAN learns the mapping $G : \mathbb{Z} \to \mathbb{X}$ such that $z_1, z_2 \in \mathbb{Z}$ which are close in the $\mathbb{Z}$ latent space produce images $x_1, x_2 \in \mathbb{X}$ which are visually similar (Xia et al., 2022). However, GANs do not offer an "inverse model", i.e., a mapping from data space back to latent space, making it difficult to infer a latent representation for a given data sample (Creswell & Bharath, 2018). Given an image, GAN inversion maps data $x \in \mathbb{X}$ back to latent representation $z^* \in \mathbb{Z}$ or, in other words, finds an image $x^*$ that can be entirely synthesized by the generator G and still remain close to the real image $x$. The inversion problem can be formulated as

$$z^* = \arg \min_z \mathcal{L}(G(z), x) \tag{1}$$

---

[1] https://pypi.org/project/tab2img/

where $\mathcal{L}(\cdot)$ is a distance metric in the image or feature space, and G is a generative feed-forward neural network.

Due to the non-convexity of $G(z)$, Equation 1 is usually a non-convex optimization problem making it hard to find accurate solutions. Therefore, many methods with formulations based on learning, optimization, or both have been developed to tackle this optimization problem. A learning-based inversion technique seeks to train an encoder network with the objective of mapping an image into a latent space. This mapping is designed to ensure that the reconstructed image, based on the latent code, closely resembles the original image. In contrast, an optimization-driven inversion approach directly tackles the objective function using back-propagation, aiming to discover a latent code that minimizes the pixel-wise reconstruction error.

A hybrid approach combines these two strategies: it initially employs an encoder to generate an initial latent code and then iteratively refines it using an optimization algorithm. It's worth noting that, in general, learning-based GAN inversion methods often struggle to faithfully reconstruct the content of the original image. Furthermore, an additional critical concern in optimization-based GAN inversion is the proper initialization of the latent code. Given that Equation 1 represents a highly non-convex problem, the quality of the reconstruction is heavily reliant on the choice of an appropriate initial value for $z$. One intuitive solution involves starting the optimization process with several random initial values and selecting the result with the lowest associated distance metric, as this can significantly impact the overall success of the inversion.

## 3 RELATED WORK

### 3.1 DEFENSE-GAN

Many defense methods against adversarial attacks are aimed at defending against a specific attack method (Samangouei et al., 2018). Samangouei et al. (2018) suggest a method called Defense-GAN which is consistently effective against different attack methods (attack-agnostic) and is effective against both white-box and black-box attacks. Defense-Gan utilises the generative capabilities and the distribution learnt from benign data of the generative model $G$ in the GAN architecture. At inference time, the generator creates an output $x'$ which is similar to the input $x$ but with minimized adversarial changes. To do this, the framework samples $R$ random initializations of $z$ from the latent space. Then it tries to find $z'$ by performing a fixed number of gradient descent (GD) steps which minimize $\|G(z) - x\|_2^2$. Once the best $z'$ is found, $x'$ is generated by applying the generator $G$. The generated $x'$ is passed on to the classifier and should be classified correctly since the generator only learnt the benign data distribution. With too many GD iterations some of the adversarial noise components may be retained resulting in decreased classification performance, hence, one must be careful when choosing the amount of GD iterations.

### 3.2 CTGAN

Modeling tabular data containing a mix of continuous and discrete features is a difficult task (Xu et al., 2019; Wang et al., 2023), since usually the discrete features are highly imbalanced and continuous features contain multiple modes. Xu et al. (2019) propose a CTGAN which uses a conditional generator to addresses several unique properties of tabular data:
**Mixed data types** - In the real world tabular data consists of mixed features meaning that GANs would need to apply both softmax for the categorical features and tanh for the continuous features on the output.
**Non-Gaussian distributions** - Continuous values in tabular data are mostly non-Gaussian and applying a min-max transformation can lead to vanishing gradients, whereas in images pixel values follow a distribution similar to that of a Gaussian which can easily be normalized applying a min-max transformation.
**Multi-modal distributions** - A vanilla GAN could not model all modes on a simple 2D dataset (Srivastava et al., 2017), hence it would also have a hard time modeling the multi-modal distribution of continuous features.
**Learning from sparse one-hot-encoded vectors** - During the training of a GAN it generates a probability distribution over categories using softmax, whereas non-synthetic data is represented with one-hot encoded vectors meaning that the discriminator could easily distinguish real and syn-

thesizes data by checking the distribution's sparseness.

**Highly imbalanced categorical columns** - In categorical features the major category appears in more than 90% of the records which can lead to severe mode collapse.

To tackle the problem of non-gaussian distributions in continuous features and since properly representing the data is critical in training neural networks, Xu et al. (2019) proposed mode specific normalization where each feature is processed independently and each value is represented as a one-hot encoded vector representing the mode and a scalar representing the value within the mode.

Xu et al. (2019) solved the problem of highly imbalanced categorical features and learning from sparse one-hot-encoded vectors by introducing the conditional vector, the training-by-sampling method and altering the generator loss. Properly sampling the conditional vector and training data can help the model evenly explore all possible values in discrete features and as the training advances adding the cross entropy loss between the conditional vector and one hot encoded vector allows the generator to make an exact copy of the given one hot encoded vector.

### 3.3 DENOISING AUTOENCODER

A Denoising Autoencoder (DAE) is a type of artificial neural network, specifically a variation of the traditional autoencoder, designed to learn efficient representations of data by removing noise or corruption from it. DAEs consist of an encoder and a decoder, just like standard autoencoders, but they are trained to map noisy or corrupted input data to clean, uncorrupted versions of that data. They find applications in various domains, including image processing, anomaly detection, feature learning, and more, where data denoising or representation learning is required.

In image proccesing, the DAEs can be trained on noisy images and taught to reform clean versions. This is useful in various applications, especially in the adversarial robustness (Meng & Chen, 2017; Sahay et al., 2019) setting. DAEs can also be used for anomaly detection (Meng & Chen, 2017). If the autoencoder is trained on a dataset of normal data, it will learn to reconstruct normal data accurately. When presented with anomalous or unusual data, the reconstruction error will be higher, indicating an anomaly.

## 4 PROPOSED METHOD

Our methods consists of: (1) Converting our tabular records into images since we are training image based GANs to utilize their capabilities. (2) Training two distinct image based GANs each on a different class (3)Identifying the classification threshold for one of the two trained GANs and by this splitting our inference into two steps. Since the GAN inversion technique is a costly one, this split allows us to perform it only once in the best case scenario saving us time during the inference. (4) Step 1: Perform GAN inversion on a single GAN and use the obtained losses and classification threshold to receive a classification. If this classification and the targets model classification on the original record agree, we are done, else, Step 2. (5) Step 2: Perform GAN inversion on the second GAN. Now the images found in the GAN inversion process are turned back into tabular records and classified along side the benign tabular record. A majority vote is performed to determine the class.

### 4.1 TRAINING PHASE

To properly train a GAN which is based on image data we first need to convert our tabular records into images. Using the Tab2Image framework we convert each tabular record in our training set into an $nxn$ image where $n$ is the height and width of the image, note that $n$ can vary across different datasets (elaborated in the evaluation section). After obtaining the converted training set of $nxn$ images, we use the PIL's libraries image resizing capabilities to resize each image to the desired size of the used GAN model. Once we obtained a training set of correctly sized images $D$ we split it into two training sets based on the true label, where images labeled as 0 go into $D_0$ and images labeled as 1 go into $D_1$. To tackle the problem of class imbalance, over-sampling could be performed to the minority class or under-sampling could be performed to the majority class. Then, two GANs are trained, each one on its respective dataset $D_0$ and $D_1$, and two generators $G_0$ and $G_1$ are extracted since in our method we have no use for the discriminators, excluding for the GANs training.

Once training is completed, the classification loss reconstruction threshold needs to be set and the generator upon which it will be effective need to be chosen. The generator is chosen and the threshold is set based on validation samples. The process which is described below is performed on both of the generators $G_1$ and $G_2$. For each sample, using a generator $G$ we perform a fixed number of gradient descent steps $S$ using $R$ different random initializations from the latent space $z \in \mathbb{Z}$ minimizing the $\mathcal{L}_2$ loss as described in Equation 1, except we extract the numerical value which the loss achieved in the optimization process and not the actual $z$. We average these losses across all initializations of $z$ for this specific sample. This average is our threshold $t$ and it is evaluated against all of the other samples thresholds. During threshold setting, the samples are classified based on the $t$ at hand. For example, if we are evaluating $G_0$ and the value is below $t$ the sample will be classified as 0 since we assume that $G_0$ will be better at minimizing the loss if the samples true value is 0, otherwise it will be 1. For the second generator $G_1$ the labels are inverted. The threshold with the highest classification accuracy is chosen and the generator which provided it will be selected to be used in the inference phase.

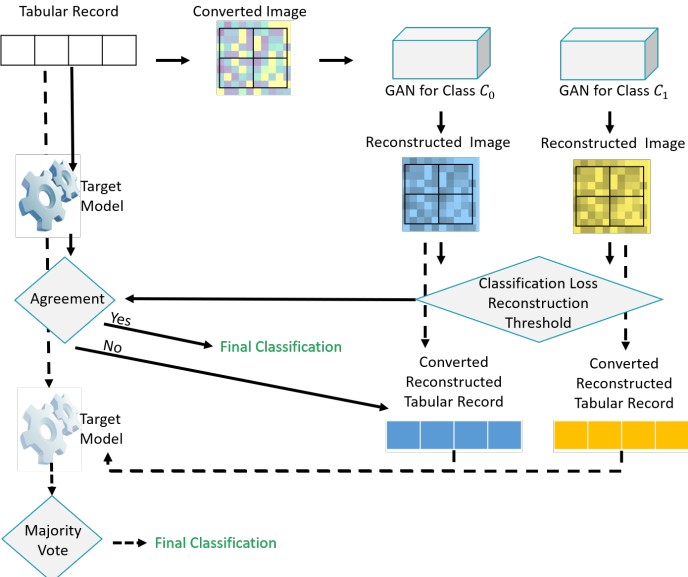

Figure 1: We show here the inference phase of our method which is divided into two steps. All the line which are bold lines are represented as being part of Step 1 and all the dashed line are represented as being part of Step 2. The reconstructed images are the images we receive after applying the generator to the latent representation $G(z)$. The Converted Reconstructed Tabular Records are the records we receive after applying resizing and inverse convert on the reconstructed images.

## 4.2 INFERENCE PHASE

During the inference phase, given a tabular data record, we first convert the record into a $nxn$ image using the Tab2Image framework, then the image is resized to the input size of the GANs models and we get image $x$. Previously in the training phase we have identified a threshold $t$ and a generator $G$ which supports this threshold. Lets say that the selected generator is $G_0$. Employing this generator we perform optimization-based GAN inversion with fixed number of gradient descent steps $S$ for a single sample using the $\mathcal{L}_2$ loss with Equation 1 for $R$ different random initializations from the latent space $z_0 \in \mathbb{Z}_0$. Now we average the acquired losses for each of the $R$ initialization of $z_0$ and check if the averaged value is above or below the selected threshold $t$. If it is below, the GANs classification for this record is 0 and if it is above the classification is 1, this could be inverted if $G_1$ was chosen as the generator with the threshold in the training phase. We now pass the original tabular record to the target model and it predicts its class. If the targets model classification matches the GANs classification based on the threshold then they agree and this is our final classification, if not we continue to the next step.

The optimization-based GAN inversion is a time costly procedure therefore, initially, we only perform it on one of the generators but if we proceed to this step we must perform it on the second generator. Hence, we perform $S$ gradient descent steps for the same sample using the $\mathcal{L}_2$ loss with Equation 1 for $R$ different random initializations from the latent space $z_1 \in \mathbb{Z}_1$. Now that we have $R$ optimized initializations of $z_0$ and $z_1$ we select the one with the minimal loss from each one of these sets to acquire $z_0^*$ and $z_1^*$ respectively. We generate the images from their latent representations such that $x_0 = G_0(z_0^*)$ and $x_1 = G_1(z_1^*)$. Both $x_0$ and $x_1$ are resized into $nxn$ images and then these images are inverse converted into tabular records. These tabular records and the original tabular record is passed to the target model for classification and a majority vote is performed to determine the final classification of the original record.

## 5 EVALUATION

### 5.1 DATASETS

(1) **The LoanD** dataset is based on the information from the Fake Irish Bank. The Fake Irish Bank is a peer to peer lending bank based in the ireland, where the bank provide funds for potential borrowers. This dataset represent which loyal customers receive loans from the Irish Fake bank and default on them based on the features that represent the the borrowers credit score. The complete Loan dataset [2]is based upon Lending Club Information. It consists of 886 thousand records and 25 features.

(2) **The NSL- KDD** dataset[3] is a intrusion detection dataset which contains network traffic records, each record representing if the network traffic contains an attack and if it does what type of attack is it. It consists of 25,191 records and 25 features. The original task of multi label classification was converted into a binary classification task representing if the record is an attack or not. The preproccesing of the dataset consisted of feature selection and class balancing.

(3) **The IoT** dataset [4] for intrusion detection systems comprises 2.5 million records and 23 statistically engineered features derived from the pcap files. Four primary features were extracted from the pcap: packet count, jitter, size of outbound packets only, and outbound and inbound packets together. For each of these four features, three or more statistical measures were computed, culminating in a total of twenty-three distinct features. These features were computed based on seven distinct statistical measures (including mean, variance, count, magnitude, radius, covariance, and correlation coefficient) over a 10-second time window. The task of this dataset closely resembles that of the NSL-KDD dataset and was similary converted into a binary classification task.

### 5.2 EXPERIMENTAL SETTINGS

All experiments were performed on CentOS Linux 7 (CORE) operating system using 24G of memory and a NVIDIA GeForce RTX 3090 Ti graphics card. The code used in the experiments was written in Python 3.8.12, scikit-learn 1.1.1, NumPy 1.19.5, and TensorFlow-GPU 2.8.0.

All of the tabular datasets were split into three parts, $80\%$ for training, $10\%$ for validation, and $10\%$ for testing: (1) Training: training the target model, and training the GANs (further split into two training sets based on labels); (2) Validation: validating the trained models, and selecting the classification threshold; and (3) Testing: evaluating the target model and generating the attacks. The fixed number of gradient decent steps during optimization based GAN inversion was set at 200 and the number of random initializations was set at 5 for all of our experiments.

Note that utilizing Tab2Img to transform the provided tabular datasets into images necessitates having a square root integer as the number of features. Consequently, we applied feature selection methods to align with this requirement. In addition, since its is required to convert our images back into tabular records after optimization, we designed an inverse convert function and added it as additional phase in the Tab2Img framework. Categorical features were encoded into numeric sequential integers, since if using the one-hot encoding representation approach, the adversarial attack might

---

[2]`https://www.kaggle.com/datasets/mrferozi/loan-data-for-dummy-bank`
[3]`https://www.unb.ca/cic/datasets/nsl.html`
[4]`https://shorturl.at/djsH1`

modify a categorical feature such as gender to be both male and female, inadvertently giving it an invalid value. Also, all of the datasets were normalized between the $(-1, 1)$ range so they would match the input requirements of the chosen GAN model.

## 5.3 TARGET MODEL AND GAN SETTINGS

The target models employed were a TensorFlow random forest classifier and a XGBoost classifier (Chen & Guestrin, 2016) with no hyper-parameter initialization except for the random state. We choose to utilize classical ML models, since non-differential learning algorithms usually obtain better performance for learning tasks with tabular data (Arik & Pfister, 2021; Lundberg & Lee, 2017; Popov et al., 2019; Grolman et al., 2022), and among those, random forest and XGBoost obtained the best performances.

DCGAN proposed by Radford et al. (2015), or Deep Convolutional Generative Adversarial Network, was employed in our evaluation for several compelling reasons. First, DCGANs have demonstrated remarkable success in generating high-quality images, making them a valuable choice when it comes to tasks involving image synthesis. Furthermore, DCGANs offer a structured and well-defined architecture, making them suitable for rigorous evaluation and comparison with other image generation techniques (Creswell & Bharath, 2018). Their architectural simplicity, relative to other complex generative models, often leads to more stable training and faster convergence, which is crucial for practical applications (Xia et al., 2022). DCGANs have also been widely adopted and benchmarked in the deep learning community, making it easier to compare our results with existing literature and establish the effectiveness of our approach.

## 5.4 ATTACK SETTINGS

When evaluating our method, two state-of-the-art attacks were performed, both were part of the Adversarial Robustness Toolbox Nicolae et al. (2018): (1) HopSkipJump attack (HSJ) Chen et al. (2020a), and (2) the boundary attack Brendel et al. (2017). Since the evaluation was performed on classical ML models and not on neural networks, gradient based optimization attacks such as FGSM, PGD, and CW could not be performed,setting us in a black-box scenario, where usually query based attacks are performed. The training set used to train the surrogate model in the black box scenario was created from the original training set by choosing samples with replacement. The process of choosing these samples was seeded with a different seed from the one used to train the target model meaning that the training sets of the surrogate model and the target models differ since in a black box scenario the attacker does not have the same training samples as those used to train the target model.

## 5.5 METHODS USED FOR COMPARISON

We evaluated the performance of our proposed method and compared the results to the results obtained by the following methods:

(1) **Defense-Gan** a Gan based attack and model agnostic robustness method that denoises the original input of an image and passes it down to the target model for classification. Since the method only takes images as input we needed to perform an enhancement in the form of converting the tabular data into images using the Tab2Image framework and resizing them to the size of the input of the GAN model. The gradient decent steps and the amount of random initializations was set similarly to our method for a fair comparison.

(2) **DAE** an autoencoder based defence and model agnostic robustness framework that employs an autoencoder, as a reformer network. Where after passing the reformer network the record is passed to the target model for classification. This method also was evaluated only on images hence, we perform an enhancement and employ a variational autoencoder fit for tabular data as the reformer network. We evaluate our method against the reformers networks denoising capabilities.

(3) **CTGAN** is not a robustness method but rather a GAN architecture that is designed specifically for tabular data. Due to the transformations that the data undergoes to fit CTGANs specifications such as one hot encoding for continuous features (as elaborated in the related work section) performing optimization based GAN inversion with a $\mathcal{L}_2$ loss function has no meaning and does not work

as intended. To be able to do optimization based GAN inversion using CTGANs, a novel custom loss function would need to be designed, which is a different research than the one explored in this paper.

| Target Model | Dataset | Test Set | Attack Success Rate | Benign Accuracy | TM Accuracy | | |
|---|---|---|---|---|---|---|---|
| | | | | | DAE | Enhanced Defense Gan (Images) | Our Method |
| Random Forest | IoT | Benign | 0% | 99.99% | 77.46% | 88.33% | **99.99%** |
| | | HopSkipJump | 100% | | 70.33% | 58.33% | **88.66%** |
| | | Boundary | 100% | | 77.66% | 66.67% | **99.33%** |
| | KDD | Benign | 0% | 99.56% | 64.68% | 80% | **99.99%** |
| | | HopSkipJump | 99.91% | | 63.33% | 79.83% | **89.33%** |
| | | Boundary | 99.91% | | 71% | 78% | **88%** |
| | LoanD | Benign | 0% | 100% | 96.32% | 86.67% | **100%** |
| | | HopSkipJump | 100% | | **88.33%** | 64.3% | 76.22% |
| | | Boundary | 100% | | 89% | 54% | **90%** |
| XGBoost | IoT | Benign | 0% | 99.99% | 77.18% | 87.9% | **98.7%** |
| | | HopSkipJump | 100% | | 74.33% | 60% | **84.66%** |
| | | Boundary | 100% | | 83% | 51% | **92%** |
| | KDD | Benign | 0% | 99.76% | 34.96% | 48.67% | **56.54%** |
| | | HopSkipJump | 99.67% | | 34.33% | 48% | **55%** |
| | | Boundary | 99% | | 36% | 47.33% | **49.41%** |
| | LoanD | Benign | 0% | 100% | 96.32% | 96% | **97.33%** |
| | | HopSkipJump | 100% | | **88%** | 59.33% | 65% |
| | | Boundary | 100% | | **87%** | 45.68% | **87%** |

Table 1: The table above shows the results of our emperical evaluation on two different models and three different datasets.

## 5.6 RESULTS

Table 1 presents the results of each one of the evaluated methods across two different target models and three different datasets. The target model column specifies which target model is being evaluated and the dataset column further indicates on which dataset the target model is being evaluated. The test set column represents which test set is being evaluated and if it is an adversarial one, also it specifies which attack was employed on the test set. The attack success rate column indicates how well the attack was at fooling the employed target model. Note that in the benign test set the attack success rate is always 0% since no attack has been performed on it. Benign accuracy indicates how well the target model performed on the classification task with the unperturbed test set i.e., when the target model is not subjected to any attack whatsoever. The TM Accuracy consist of three columns each showing the accuracy of the target model after a specified robustness method was applied on the test sets using: (1) the enhanced Defence-Gan method; (2) a denoising (reformer) autoencoder (DAE) and, (3) our proposed method.

As can be seen from Table 1 the accuracies of both of the examined target models: (1) Random Forest and (2) XGBoost on a benign test set are close to 100% classification accuracy. In addition, both adversarial attacks HopSkipJump and Boundary are performing well since across all dataset and target models they reach attack success rates close to 100%. In general it can be seen that our proposed method achieves high results since across most datasets and across different target models if outperforms (in bold) the other examined methods (Defense-Gan and DAE).

When examining the results for the IoT and LoanD datasets we note that in the benign test case Defense Gan outperforms the DAE method in the IoT dataset. However in the LoanD dataset the DAE outperforms Defense Gan. Note, that in the IoT dataset Defense Gan outperforms the DAE in more than 10% in both examined target models. Whereas in the LoanD datasets the DAE outperformed

Defense Gan by 10% only in the RF target model. When the target model is based XGBoost they reach the same results (96%) in the LoanD dataset.

In addition, for the IoT and LoanD datasets, after the test set undergoes an attack the DAE method performs better than Defence Gan with a maximum improvement of 41% compared to Defence Gan and a minimum improvement of 10%. This could probably be explained by the fact that the DAE performs the reconstruction directly on the tabular data, whereas, the Defense Gan must reconstruct data that has been converted into images. Note that, our method also has to work on data that has been converted into images but the separation of the training data into each class and training separate GANs for each class allows our method to better distinguish the difference between the classes compared to defense GAN where a single GAN is trained for both the classes.

When focusing solely on the LoanD dataset, it can be seen that the DAE outperforms our method in two scenarios, however in both of them it is on a test set which was perturbed by the HopSkipJump attack. This could probably be explained by the fact that HopSkipJump performs slight perturbations which are close to the decision boundary hence for a DAE that performs directly on tabular data it must be easier to cross the boundary and bring the record back into its original class.

When examining the results of the KDD dataset, it can be noticed that even thought the XGBoost target model achieves the same results as the Random Forest model in the benign test case scenario, the results are not that promising after applying any defense method. This could be probably explained by the fact that random forests typically outperform gradient boosting in high noise settings, especially with small data (which in this case is true). Gradient boosting aims to reduce bias (systematic errors) by sequentially fitting weak learners to the residuals of previous learners. While this can lead to very accurate models, it can also make the algorithm more sensitive to noise and outliers. Random forests, on the other hand, tend to have higher bias but lower variance. In high noise settings such as adversarial samples, reducing variance is often more critical than reducing bias.

## 5.7 DISCUSSION

Using GAN based defence mechanisms against adversarial attacks in the image domain has proven to be effective in enhancing the robustness of machine learning models (Samangouei et al., 2018; Chen et al., 2020b). GANs usually assume that the data they generate follows a continuous probability distribution. Tabular data often contains discrete values or categorical variables, which do not fit the continuous distribution assumptions of GANs making it hard for GANs to synthesize tabular data (Xu et al., 2019). Further, GANs work well when the data has a high dimensionality, but tabular data typically has a lower dimensionality, which can make it challenging for GANs to capture the underlying patterns effectively.

In order to bypass these challenges and to utilize the effectiveness of GANs we tried to convert our tabular data into image data and resize it to the right dimensionality. However, after doing so and applying the Defence GAN method to the data we note that there was still a disparity of over 10% between a benign test set and an adversarially perturbed one. Understanding that on the one hand, GANs are designed to generate data that has a certain underlying structure, such as the spatial structure in images or the sequential structure in text, but tabular data, on the other hand, lacks such inherent structure, we decided to split the training data into classes and train a GAN for each one of them, in a way providing them with some kind of structure. After performing this and with the combination of resizing and converting the data to images we were finally able to achieve results which are close to the benign classification of the target models.

## 6 CONCLUSION AND FUTURE WORKS

In this paper we presented a novel framework which employs a image conversion technique and GAN models generative capabilities to boost the robustness of classical ML models against adversarial examples. Our method shows promising results compared to the evaluated methods. Future work could include extending our method to multi-label classification problems and adapting the GAN inversion technique directly to the CTGAN.

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
