# OpenReview forum: "Tab2Gan: Utilizing image conversion and Gan inversion for tabular model robustness"
_ICLR.cc/2024/Conference — ICLR 2024 Conference Withdrawn Submission_

### Official Review · Reviewer_PeVY · 2023-10-25

**Soundness:** 3 good
**Presentation:** 2 fair
**Contribution:** 2 fair
**Rating:** 5
**Confidence:** 4

**Summary:**

This paper proposes a novel attack-agnostic robustness method that utilizes the generative capabilities of image-based GANs to enhance the robustness of classical machine learning models trained on structured (tabular) data. The authors employ two GANs, each trained on benign data from different classes, to classify incoming inputs and reconstruct them within the benign data distribution before presenting them to the target model. The proposed approach consistently outperforms existing techniques in terms of classification accuracy on three structured datasets and two machine learning models.

**Strengths:**

(1) The paper introduces a novel approach that leverages GANs to enhance the robustness of machine learning models applied to tabular data. This approach is attack-agnostic and shows promising results in terms of classification accuracy.

(2) The paper provides a thorough description of the proposed method, including the framework architecture, training of GANs, and evaluation process. The details allow the reader to understand the methodology and reproduce the experiments.

(3) The paper discusses the background of Tab2Img and image resizing techniques, as well as the concept of GANs and their inversion.

**Weaknesses:**

(1) The paper lacks comparisons and discussions with widely-known baselines in the field. While it claims to outperform existing techniques, there is no direct comparison presented in the paper to support this claim.

(2) The paper only evaluates the proposed method on three structured datasets and two machine learning models. A broader evaluation on more diverse datasets and models would strengthen the generalizability of the proposed approach.

(3) The paper could benefit from providing more implementation details and ablation studies to further analyze the performance and effectiveness of the proposed method.

**Questions:**

There are many transfer-based adversarial attacks based on alternative networks. How does your defense perform against these attacks?

**Details Of Ethics Concerns:**

/NA

---

### Official Review · Reviewer_tf3D · 2023-10-31

**Soundness:** 2 fair
**Presentation:** 2 fair
**Contribution:** 2 fair
**Rating:** 3
**Confidence:** 4

**Summary:**

This paper proposes a novel attack-agnostic robustness method that utilizes the generative capabilities of image based generative adversarial networks (GANS) to enhance the robustness of classical machine learning models trained on structured (tabular) data. To safeguard the target models, the authors employ two GANs, each trained on benign data from different classes. The evaluations are conducted on three commonly known structured datasets and two conventional machine learning models, the proposed robustness approach consistently outperformed the existing techniques in the field.

**Strengths:**

1. This paper introduces an innovative and attack-agnostic approach to enhance the robustness of classical machine learning models when applied to structured (tabular) data. The proposed method represents a novel paradigm in defending against adversarial attacks in domains typically less explored by GAN-based defenses.
2. The proposed approach of this paper capitalizes on the versatility of GANs to not only detect adversarial inputs but also reconstruct them within the benign data distribution, effectively mitigating the impact of potential attacks. This paper also provides a comprehensive description of the proposed attack-agnostic robustness method.

**Weaknesses:**

1. Although this paper proposes a new approach for the robustness on the tabular data, the motivation of the proposed approach is still not that clear, such as why to utilize and combine these of the two GANs, and what rules this paper use to choose the GANs for the conversion. The influence of the classification loss on the robustness should also be discussed.
2. This paper shows the inference phase of the proposed method in the Figure 1 with two steps. Yet, it would be more sufficient to clarify the influence of converted image and reconstructed image in the section of experiments with more empirical evidence such as some visualization results in the process of generation. The analysis of choosing hyperparameters in the training process of this framework should also be provided in the experiments.
3. This paper mentioned that the GAN inversion technique is a costly one and performed only once in the best-case scenario for saving time during the inference. Thus, it would be more persuasive for this paper to provide the comparison of the computational cost between the proposed approach with GAN inversion and its counterparts.

**Questions:**

This paper proposes an approach to improve the robustness of conventional ML model on the tabular data by using the GAN inversion. Obviously, it still needs further details with evidence and explanations to clarify the effectiveness of the proposed method.

---

### Official Review · Reviewer_CeqD · 2023-11-01

**Soundness:** 2 fair
**Presentation:** 2 fair
**Contribution:** 2 fair
**Rating:** 3
**Confidence:** 3

**Summary:**

This work studies the defense method for tabular data models using image conversion and GAN inversion. The proposed method consists of two GANs, both of which were trained on benign data, to discriminate benign from adversarial ones. The method is easy to understand.

**Strengths:**

- The defense against tabular data is an important topic and less studied(compared with AML in CV, NLP area).
- Experiments were conducted to show its effectiveness as an adversarial defense.

**Weaknesses:**

- Defenses using GANs have existed since 2018, which hinders the technical novelty of this work.
- An effective defense should be evaluated with adaptive attacks (e.g. [1]) to convincingly show its practical effectiveness. This part should be analyzed and included.
- The conclusion part has been overclaimed - the scope is only for tabular data models rather than to boost the robustness of classical ML models.
- Some analysis is supposed to be included to show the effectiveness of the paper.
- The presentation can be further improved.

References.
[1] Tramer, Florian, et al. "On adaptive attacks to adversarial example defenses." Advances in neural information processing systems 33 (2020): 1633-1645.

**Questions:**

See weakness.

**Details Of Ethics Concerns:**

NA.

---

### Official Review · Reviewer_pzJ6 · 2023-11-01

**Soundness:** 2 fair
**Presentation:** 1 poor
**Contribution:** 2 fair
**Rating:** 3
**Confidence:** 5

**Summary:**

This paper proposes a novel approach to enhancing the robustness of machine learning models trained on structured data against adversarial attacks. The approach utilizes Tab2Img to convert the tabular data to image data, leverages image-based generative adversarial networks (GANs) to detect and reconstruct adversarial inputs within the benign data distribution. The proposed method outperformed existing techniques in the field, resulting in classification accuracy that closely align with the benign accuracy rate. The approach is evaluated on 3 datasets, including the LoanD, NSL-KDD, IoT, across Random forest and XGBoost model, achieving promising results than DAE and the enhaced defense GAN.

**Strengths:**

1. The research topic exploring adversarial robustness on tabular data is interesting.

2. The approach utilizes the generative capabilities of GANs to not only detect adversarial inputs but also reconstruct them within the benign data distribution is new.

3. The proposed method is competitive when compared to existing techniques.

**Weaknesses:**

1. The method is described textually in Section 4, making it complex and unclear. It would be better to Introduce algorithmic representation for clarity and reorganize content for coherence, ensuring essential details are correctly placed within the related work section in stead of the experiment part.

2. Absence of ablation studies to validate design choices and hyperparameter impacts. The authors are encouraged to conduct ablation studies to analyze the effects of different components and hyperparameter variations, enhancing the paper’s validity.

3. The paper lacks a thorough comparative analysis to justify the proposed method’s superiority. it would be helpful to provide a comprehensive analysis, emphasizing the method’s advantages over existing approaches with experimental support.

**Questions:**

NA